# Structural Insights and Development of LRRK2 Inhibitors for Parkinson’s Disease in the Last Decade

**DOI:** 10.3390/genes13081426

**Published:** 2022-08-11

**Authors:** Gunjan Thakur, Vikas Kumar, Keun Woo Lee, Chungkil Won

**Affiliations:** 1Institute of Animal Medicine, College of Veterinary Medicine, Gyeongsang National University, Jinju 52828, Korea; 2Division of Life Sciences, Department of Bio & Medical Big Data (BK4 Program), Research Institute of Natural Science (RINS), Gyeongsang National University (GNU), 501 Jinju-daero, Jinju 52828, Korea

**Keywords:** G2019S mutation, kinase inhibitors, LRRK2, neurodegeneration, Parkinson’s disease

## Abstract

Parkinson’s disease (PD) is the second most prevalent neurodegenerative disease, characterized by the specific loss of dopaminergic neurons in the midbrain. The pathophysiology of PD is likely caused by a variety of environmental and hereditary factors. Many single-gene mutations have been linked to this disease, but a significant number of studies indicate that mutations in the gene encoding leucine-rich repeat kinase 2 (LRRK2) are a potential therapeutic target for both sporadic and familial forms of PD. Consequently, the identification of potential LRRK2 inhibitors has been the focus of drug discovery. Various investigations have been conducted in academic and industrial organizations to investigate the mechanism of LRRK2 in PD and further develop its inhibitors. This review summarizes the role of LRRK2 in PD and its structural details, especially the kinase domain. Furthermore, we reviewed in vitro and in vivo findings of selected inhibitors reported to date against wild-type and mutant versions of the LRRK2 kinase domain as well as the current trends researchers are employing in the development of LRRK2 inhibitors.

## 1. Introduction

Parkinson’s disease (PD) is a neurodegenerative movement disorder affecting the elderly person. In 1817, James Parkinson clinically described PD for the first time in his report “essay on the shaking palsy” [1]. PD is the second most prevalent neurodegenerative disorder after Alzheimer’s disease, affecting approximately 10 million people globally. There is currently no cure for PD, and dopamine replacement using l-dehydroxyphenylalanine (l-DOPA) remains the primary treatment. l-DOPA is quite useful in lowering motor symptoms; nonetheless, there are two key issues: adverse effects and patients becoming medication-resistant [2]. Moreover, it is critical to emphasize that current PD therapies only reduce symptoms and do not slow or stop disease progression. The need for novel PD treatment options is obvious. Furthermore, as the world’s population ages, it is apparent that this need is increasing. PD affects 1% of the people aged more than 60 years and 5% of people aged more than 85 years [3]. PD is mainly divided into two types: familial and idiopathic. Merely around 5–15% of PD cases have an ancestral background, whereas the remaining cases are of the idiopathic type [4,5]. In post-mortem brain tissue, the typical characteristics of PD include loss of dopaminergic neurons in the substantia nigra, which is accompanied by the development of fibrillar aggregates made of α-synuclein and other proteins (e.g., Lewy bodies) [6]. PD patients often have four motor symptoms, including resting tremors, muscular stiffness, postural instability, and bradykinesia. Sleep disruption, loss of olfaction, urine problems, and constipation are among the non-motor symptoms of PD. Furthermore, other non-motor signs emerge later in the disease, including dementia, cognitive impairment, and hallucinations [7]. According to recent studies, there are presently no specific tests or assays in medical usage that might permit an early diagnosis of the disease before symptoms arise despite the fact that a number of signs might suggest the existence of PD [8]. In recent years, much study has been dedicated to the genetic origins of PD, with the hope that they would guide the way to novel therapeutic targets. The complex clinical aspects suggest that various neurophysiological pathways might influence PD pathogenesis [9,10,11]. Although the exact mechanisms underlying PD remain a mystery, recent clinical research has revealed that a number of proteins, such as PTEN (phosphatase and tensin homolog)-prompted putative kinase 1 (PINK1), α-synuclein (SNCA), VPS35, glucocerebrosidase (GBA), Parkin (PRKN), DJ-1, and leucine-rich repeat kinase 2 (LRRK2), are linked to the progression of PD [12,13]. Variations in the aforementioned genes are the most prevalent causes of familial PD. The A53T autosomal dominant mutation in α-synuclein was the first to be linked to PD [14,15]. Thereafter, many other mutations were identified in the abovementioned genes. They were reported to cause PD significantly when compared to normal gene expression. It was observed that autosomal recessive mutations are rarely observed in PD patients, whereas autosomal dominant mutations are the most prevalent [14,16]. However, a single unifying cause of PD has yet to be established at the cellular level [10,17]. The mutation in LRRK2 is the most common source of familial PD cases and is a substantial risk factor for idiopathic PD cases among the PD genes [18,19]. Due to the active involvement of this protein in PD, several LRRK2 inhibitors have been identified; however, many lack selectivity or the ability to cross the blood–brain barrier [20,21]. This review discusses recent advancements in LRRK2 structure determination, function, and the potential selective inhibitors identified in the past decade, concisely summarizing the relevant information.

## 2. LRRK2 and Its Function

LRRK2 is a large, multidomain serine–threonine kinase of 2527 amino acids (280 kDa), consisting of seven domains with distinct functions. The N to C terminal sequence of LRRK2 comprises the armadillo repeat (ARM), ankyrin (ANK), leucine-rich repeat (LRR), Ras of complex proteins (ROC), C-terminal of Roc (COR), kinase (KIN), and WD40 domains (Figure 1A) [22,23,24]. The ARM, ANK, LRR, and WD40 domains are observed in several protein–protein interactions (PPIs), whereas the ROC-COR and kinase domains are involved in two enzymatic functions, GTPase and kinase, making LRRK2 a multifunctional protein [25,26]. LRRK2 belongs to the ROCO family, with the most remarkable overall sequence similarity to mammalian paralogue LRRK1 [27]. However, LRRK2 and LRRK1 are not close homologs for the kinase domain [28]. LRRK2 protein is expressed in various brain regions and several other organs, such as the lungs and kidneys. In situ hybridization in mice demonstrated that it is highly expressed in the basal ganglia region associated with motor function in PD and non-motor regions, for example, the hippocampus [20,25,29]. LRRK2 was also found in high concentrations in leukocytes, such as monocytes, B lymphocytes, and dendritic cells [30,31,32]. Inside the cell, LRRK2 is mainly present in the cytoplasm, where it interacts with the 14-3-3 adaptor protein in a phosphorylation-dependent way [33,34]. LRRK2 is mostly inactive in the cytoplasm, where it interacts with the 14-3-3 adapter protein in a phosphorylation-dependent manner. LRRK2 has been seen to assemble into inclusion bodies when 14-3-3 binding is disturbed, indicating that 14-3-3 may stabilize normal LRRK2 folding in the cytoplasm [33,35]. Overexpressed LRRK2 forms membrane-associated dimers in a small percentage, and this dimerization is reported to boost the specific activity of LRRK2’s kinase [34,36]. LRRK2 plays an essential role in the endolysosomal system and the trans-Golgi network [37]. The function of LRRK2 in vesicular trafficking is mediated by Rab29, Rab8A, and Rab10 phosphorylation at conserved sites. A phosphoproteomics-based approach identified a subset of Rab GTPases as key LRRK2 substrates both in vitro and in vivo [38]. A total of 14 Rab members were reported to be phosphorylated by LRRK2. RILPL1, the major ciliogenesis regulator, interacts with LRRK2-phosphorylated Rab8A (Thr72) and Rab10 (Thr73), whereas RILPL2 binds to phosphorylated Rab8A, Rab10, and Rab12 [39,40]. This reveals an unanticipated impact of LRRK2 on ciliogenesis through the phosphorylation of Rab GTPases. Moreover, several studies indicate that the LRRK2 can affect the ubiquitin–proteasome system, the autophagy–lysosomal pathway, microtubule dynamics, α-synuclein phosphorylation, and immune system cells [36,41,42,43,44,45,46]. LRRK2 also participates in the protein synthesis process via interacting with 4E-BP and Ago2 proteins [47,48]. It has been postulated that LRRK2 may perform various activities in different tissues based on the subset of downstream Rab substrates expressed. Interactions of LRRK2 with important PD-related genes such as PINK1, Parkin, α-synuclein, and tau were also reported [37]. LRRK2-mediated Rab10 phosphorylation is thought to control the lysosomal enzyme β-glucocerebrosidase, whose impairment causes Gaucher’s disease [49]. Rab10 phosphorylation also affects Parkin- and PINK1-mediated mitophagy [50]. Mutant LRRK2 can induce aggregation of α-synuclein and tau protein, but the detailed mechanism involved needs further investigation [51,52,53,54]. Despite several attempts, the specific physiological role of LRRK2 remains unclear. The signaling cascade of the LRRK2 protein established by various reports indicates that LRRK2 can interact with multiple pathways responsible for the aforementioned functions [18,37].

## 3. Structural Biology of LRRK2

Despite the fact that revealing the structure of LRRK2 has been the focus of several researchers over the past decade, the outcomes achieved thus far are from perfect. The biggest stumbling block is the purified protein’s stability and expression system [55]. Many efforts have been attempted, but no desirable X-ray structures have been obtained for full-length LRRK2. However, individual X-ray structures for the ROC and WD40 domains were successfully generated and are available publicly (https://www.rcsb.org/, accessed on 10 February 2022). Several full-length cryo-EM structures were solved for the LRRK2 protein. Figure 1B shows the recently solved cryo-EM structure that was reported for the LRRK2 protein (PDB: 6VNO) [56]. The full-length structure identified using cryo-EM revealed that LRRK2 adopts a J-shaped structure in 3D space. The multidomain protein is characterized by a 200–250 amino acid ROC (Ras of complex protein) domain followed by a 300–400 amino acid domain called COR. The ROC area chiefly intervenes in the GTPase function of LRRK2, a district typical for the Ras superfamily of GTPases that goes about as a sub-atomic switch to control kinases through guanine nucleotide-subordinate conformational changes. Myasnikov et al. further revealed the interaction between each domain of the LRRK2 domain and highlighted the key residues responsible for interactions. The COR-ROC interaction site is near four high-risk mutations: Asn1437, Arg1441, Tyr1699, and Ser1761, indicating the role of ROC-COR coupling in PD [24].

The cryo-electron microscopy (Cryo-EM) LRRK2 structure reveals that the kinase domain of the protein ranged from amino acids 1878 to 2138. The LRRK2 kinase domain in this structure is reported in an inactive open state conformation (Figure 1A,B) [56]. The kinases are a class of enzymes responsible for the transfer of γ phosphate from nucleotide triphosphates (typically ATP) to one or more amino acid residues on a protein substrate side chain, resulting in a conformational shift that affects protein function [57,58]. Protein kinases are classified into three groups: serine/threonine, tyrosine, and dual-specificity based on substrate selectivity. The LRRK2 kinase domain is identified as the serine/threonine type. The sequence of MLK1 and mitogen-activated protein kinase 9 (MKKK9) have substantial sequence homology in LRRK2’s kinase domain. The majority of human kinases have a similar structural arrangement composed of an N-terminal lobe made up of five β-sheet strands and α-helix, known as the C-helix, and a C-terminal lobe made up of eight α-helices and four short β-sheet stands, which are responsible for substrate phosphorylation [59]. A similar structural arrangement was observed in the LRRK2 kinase 3D-conformation (Figure 1C) [56]. The active site of the LRRK2 kinase domain is surrounded by a glycine-rich loop (G-loop, 1886–1893 residues), a hinge region that connects the N and C terminal lobes (1947–1951 residues), αC helix (1915–1925 residues), and DYG motif (2017–2019 residues), which is the part of the activation loop (2017–2042) [60]. The activation loop reported in the original structure was disordered from residues 2020–2035; however, it was gap-filled using Discovery studio (Figure 1C). Most kinases have a highly conserved ATP binding site and catalytic amino acid residues involved in phosphorylation activities. Generally, kinases switch between catalytically active ATP-binding conformations and inactive conformations in response to phosphorylation [61,62,63,64]. The movement of an ‘activation loop’, which comprises three amino acid residues, aspartate, phenylalanine (or tyrosine in the case of LRRK2), and glycine, causes this conformational change. The LRRK2 kinase domain is an example of an unusual kinase, where the conserved DFG motif is replaced with DYG [56,60]. The knowledge of the active site of the enzyme is the basis for structure-based drug development. Previous structure-based studies used homology-based approaches to show the molecular details of the kinase domain [25,65,66,67,68]. Figure 2A shows the recently published Cryo-EM structure of LRRK2 reported in the mutant G2019S form bound with ATP (PDB: 7LI3) [24]. The detailed intermolecular interactions between the kinase domain of LRRK2 and the ATP molecule are shown in Figure 2B,C. The structural analysis reveals that the G-loop and hinge region residues are involved significantly with ATP. The ATP molecule forms two hydrogen bonds with hinge region residues Glu1948 and Ala1950.

Interestingly, in a parallel study, the binding mechanism of Type I LRRK2 G2019S inhibitors was revealed using molecular docking and molecular dynamics simulations following similar interactions with active-site residues. The modeling results demonstrate that hinge region residues Glu1948 and Ala1950 are critical in maintaining the intermolecular hydrogen bond interaction with the G2019S LRRK2 protein and inhibitor [70]. The gatekeeper residue Met1947 connects the N and C terminals of the kinase to form pi-sulfur interactions with the adenine ring of ATP. The G-loop residues Val1893, Ala1904, and Leu2001 form pi-alkyl interactions with ATP. The residues involved Van der Waals interactions for Ser1892, His1998, Gly1953, Leu1949, Ile1933, Gly1888, and Gly1891. Lys1906 and Asp1887 interact via attractive charge. Ser1954 forms a carbon bond interaction with the oxygen atom of the ATP molecule. The role of Ile1933 was recently linked with R and C spine interactions; therefore, interaction with this residue is desirable during inhibitor design. The regulatory triad of the LRRK2 kinase domains includes Lys1906, Glu1920, and DFG motif residue Asp2017 [60]. The ATP molecule interacts with Lys1906 and not with other important residues. None of the kinase domain–mutated residues forms interactions with the ATP molecule similar to the previous homology obtained structures [65,68,70]. The knowledge of the intermolecular interactions between the G2019S LRRK2 kinase domain and ATP may be critical for the development of G2019S-specific inhibitors over WT LRRK2 protein.

## 4. LRRK2 Mutations

LRRK2 mutations are the most prevalent causes of familial PD [71,72,73]. LRRK2 mutations are also related to various disorders such as Crohn’s disease, ulcerative colitis, leprosy, and cancer [74,75,76,77,78,79]. To date, over 80 mutations in LRRK2 have been reported; however, the role of 8 mutations in PD was significantly highlighted in many reports [80]. These mutations are reported in different domains of LRRK2, such as the ROC domains (N1437H, R1441G/H/C), COR domain (Y1699C), and KIN domains (I2012T, G2019S, and I2020T) [81]. Furthermore, other variations, such as G2385R, R1682P, and S1761R, have been discovered in PD patients and are considered to be pathogenic [82,83,84]. The recent study by Simpson et al. reveals the prevalence of 10 LRRK2 mutations in PD [85]. There are multiple additional variations associated with PD risk, including A419V, N551K, R1325Q, R1398H, T1410M, R1628P, M1646T, S1647T, N2081D etc. [74,77,86,87,88,89]. It is currently unknown to what extent these risk mutations influence the kinase activity of LRRK2 and require further investigations. The substitution G2019S is the most often found mutation in PD patients, followed by the R1441G/H/C ROC domain mutation. The kinase domain mutation G2019S is not only reported in familial PD but also reported to contribute to 1–5% of sporadic PD cases [85]. It is noteworthy to mention that PD-associated mutations have been observed with different incidence and prevalence rates in different ethnic populations. For example, the most common G2019S mutation was observed significantly in the North African population (40%), followed by the Ashkenazi Jewish (28%) and European populations (6%) [80,90]. However, the mutation is rarely observed in Asian populations [91,92,93]. The second most kinase domain mutation associated with PD, I2020T, was observed in European and Asian populations but with lower frequencies [85]. The remaining kinase domain, I2012T, has only been reported in the Taiwanese population [94]. The G2019S mutation occurs inside the LRRK2 kinase domain’s conserved “DYG” motif. This mutation was reported to have a modulatory role in the kinase activity of the protein. According to several studies, G2019S mutations enhance the kinase activity by two- to three-fold by increasing the catalysis rate and not enhancing the substrate affinity [95]. Similarly, other pathogenic PD mutations, such as R1441C/G/H, Y1699C, and I2020T, were also reported to moderately enhance the kinase activity of the protein [95]. The pathogenic LRRK2 PD mutations discussed above interfere with the modulation of LRRK2 cellular phosphorylation sites. In the G2019S mutation, Ser910, Ser935, Ser955, and Ser973 are completely phosphorylated, whereas these sites are hypo-phosphorylated in the presence of R1441G/C, Y1699C, and I2020T mutations [95,96].

## 5. Activation of the Kinase Domain

The monomeric form of LRRK2 has been shown to be predominantly cytosolic in cells, whereas the homodimeric form links with biological membranes. The phosphorylation of residues S910 and S935, situated between LRRK2’s ankyrin and LRR domains on the N-terminal region of LRRK2, by PKA facilitates LRRK2’s interaction with 14-3-3 in the cytoplasm [97]. Further research indicates that several LRRK2 residues, including S1403, T1404, T1410, S1444, and T1491 in the ROC domain and T1967, T1969, T2031, S2032, and T2035 in the kinase domain, are potential autophosphorylation sites [95]. In vivo experiments revealed autophosphorylation at S1292, with substantial phosphorylation reported in LRRK2 mutants [98]. The phosphorylation of serine residues mainly (S910, S935, S1292, T1410, T1503, T1969) is indirectly connected to LRRK2 kinase activity, and LRRK2 kinase inhibitors induce dephosphorylation of the aforementioned serine residues [95]. As a result, serine phosphorylation and dephosphorylation are employed to track LRRK2 activity in cells. According to emerging evidence, the GTP-bound Rab family of GTPases appears to be the primary membrane recruiting agents. A subset of Rab proteins can connect to the N-terminus of LRRK2 and cause it to be localized to specific membrane organelles, depending on the Rab isoform [55,99]. However, additional research remains necessary to determine the precise mechanism of Rab protein function. The membrane recruitment caused GTP hydrolysis and subsequent dimerization, which led to kinase activation (Figure 3) [26,100]. The LRRK2 dimer exhibits higher kinase and GTPase response activity than the LRRK2 monomer [26]. The lower kinase activity of the GTP-binding mutant T1348N indicates that GTP binding is needed for LRRK2 protein kinase activity [101,102]. Pathogenic mutations in the ROC and COR domains have been shown to change GTP binding, and hence play an important role in maintaining the active site of the kinase domain. Surprisingly, this is not applicable to pathogenic mutations in the kinase domain [80,103]. The kinase domain mutations, namely G2019S and I2020T, have been shown to phosphorylate a variety of proteins, including mitogen protein kinases 3, 4, 6, and 7, resulting in the activation of many downstream signaling cascades [104,105]. Activated LRRK2 kinase is also known to phosphorylate a number of Rab family members via Rab29 and VPS35, which ultimately helps in LRRK2 re-localization, endosomal trafficking, synuclein secretion, and centrosomal defect. Unregulated expression of wild-type and PD-associated mutant variants of LRRK2 causes neuroinflammation, neuroapoptosis, and neurodegeneration in PD [37,80,99]. Thus, targeting the kinase domain LRRK2 is a promising method for treating familial and sporadic PD [20]. Considerable attempts have been undertaken in recent decades to discover LRRK2 kinase inhibitors, but little progress has been obtained. The next part goes through the major contributions to kinase inhibitor design.

## 6. Kinase Inhibitors

The dysregulated, overexpressed, or mutated protein kinases are seen in many diseases, including cancer and inflammatory and neurological disorders, and have been widely studied targets for the development of novel medicines over the last two decades. The FDA presently approves 68 kinase inhibitors (KIs) (FDA, 2021), with over hundreds of likely inhibitors in various stages of clinical studies globally [106]. Kinase inhibition is a commonly used treatment approach for neurological disorders [107,108]. Several methods for evaluating and fine-tuning kinase inhibitors have already been developed. Since enhanced kinase activity appears to be a key factor in the progression of familial PD [18] from a pharmacological standpoint, this is the most straightforward strategy to address LRRK2 clinically in all PD-causing mutations. Inhibiting LRRK2’s kinase domain has been shown to have neuroprotective effects and avoid endolysosomal deficits. Several reports have identified a number of LRRK2 kinases and described them in detail [61,109]. However, in the present report, we only considered commercially available LRRK2 kinase inhibitors (Table 1). A literature survey reveals major progress made in the field of ATP competitive Type I inhibitors; however, a few Type II inhibitors were also reported but have not been found very effective to date [110,111].

LRRK2-IN-1 was the foremost selective LRRK2 inhibitor which was identified by screening 300 compounds designed to target the ATP-binding site. Compound LRRK2-IN-1 demonstrated the greatest affinity for wild-type and G2019S, with IC_50_ values of 13 nM and 6 nM, respectively, among a variety of benzodiazepine scaffolds [112]. Furthermore, compounds had a lower affinity for A2016T and G2019S + A2016T, with IC_50_ values of 2450 and 3080 nM, respectively. LRRK2-IN-1 has a high selectivity potential and only showed affinity for 12 kinases. In HEK293 cells, human blastoid cells, and SHSY5Y cells, the inhibitor was able to induce dephosphorylation at S190 and S935. However, it failed to do so for A2016T and double mutant G2019S + A2016T. LRRK2-IN-1 displayed favorable pharmacokinetic properties in mice. Intraperitoneal injection with 100 mg/kg displayed complete dephosphorylation at S910 and S935 in the kidney but failed to do so inside the brain [112].

Utilizing a chemo-proteomics approach, the LRRK2 inhibitors CZC-25146 and CZC-54252 were generated. In a time-resolved fluorescence resonance energy transfer (TR-FRET) assay, CZC-25146 displayed IC_50_ values of 4.76 and 6.87 nM for wild types and G2019S, respectively, whereas CZC-54252 had IC_50_ values of 1.28 and 1.85 nM. respectively [113]. Kinase selectivity findings indicated that CZC-54252 can target 10 kinases out of 184, while CZC-25146 can only target 5. CZC-25146 was also shown to be non-cytotoxic in human neurons and to have a favorable pharmacokinetic profile. Unfortunately, both compounds had weak brain penetration and were thus restricted to in vitro research [113]. Choi et al. discovered the brain penetrant LRRK2 inhibitor HG-10-102-01 in 2012 utilizing an aminopyrimidine-based scaffold [114]. In a biochemical study, the compound inhibits wild type, G2019S, A2016T, and G2019S + A2016T double mutants with IC_50_ values of 20.3, 3.2, 153.7, and 95.9 nM, respectively [114]. In HEK293 cells, dose-dependent suppression of S910 and S935 phosphorylation was detected at 1 and 0.3 μM for wild type and G2019S, respectively, while 1–3 μM doses were effective against the G2019S + A2016T double mutant and A2016T. Furthermore, in wild type and G2019S, similar dose-dependent effects were seen in human lymphoblastoid, mouse Swiss3T3, and fibroblast cells. The compound’s pharmacokinetic profile displayed high oral bioavailability, a short half-life, and minimal plasma exposure. The pharmacodynamics analysis of HG-10-102-01 showed that a 50 mg/kg dosage nearly completely inhibited phosphorylation in all organs, including the brain. HG-10-102-01 showed remarkable selectivity against 138 recombinant kinases, KinomeScan (451 kinases), and KiNativ profiling. The optimization of HG-10-102-01 resulted in the development of GNE-7915, a highly selective brain penetrant inhibitor [118]. GNE-7915 had a biochemical Ki of 1 nM and a cellular Ki of 9 nM against pLRRK2. Invitrogen kinase profiling with 187 kinases yielded only TTK inhibition higher than 50%. GNE-7915 demonstrated outstanding pharmacokinetic qualities both in vitro and in vivo. GNE-7915 pharmacodynamics were studied in BAC transgenic mice by expressing human LRRK2 with G2019S. Furthermore, the PK of GNE-7915 was tested using cynomolgus monkeys, and the results show that the compound may be explored in preclinical studies. GSK2578215A, a strong and selective benzamide scaffold–based inhibitor, was discovered in 2012 [115]. GSK2578215A had biochemical IC_50_ of 10.9 and 8.9 nM against wild-type LRRK2 and G2019S, respectively. It also demonstrated modest nanomolar activity against the A2016T and G2019S + A2016T mutants, with values of 81.1 and 61.3 nM, respectively. GSK2578215A showed extraordinary selectivity for LRRK2 across the kinome, inhibiting smMLCK, ALK, and FLT3 exclusively. GSK inhibited S910 and S935 phosphorylation of both wild-type LRRK2 and the G2019S mutant in HEK293 cells at 0.3–1.0 μM. Furthermore, pharmacokinetic investigation in mice demonstrated that after intraperitoneal treatment of 100 mg/kg, the compound effectively inhibited S910 and S935 phosphorylation in the kidney and spleen but not in the brain [115]. Hermanson et al. developed a high TR-FRET cellular assay for pS935 LRRK2 and found 16 possible hit compounds after screening 1120 physiologically active compounds against LRRK2-GFP G2019S transduced SHSY5Y cells [117]. The IKK-16 (IKK Inhibitor VII) is an inhibitor of IB kinase (IKK), with an IC_50_ value of 40 nM and is one of the hit compounds that inhibited LRRK2 with an IC_50_ value of 50 nM in a biochemical experiment. IKK-16 is orally accessible in rats and mice, demonstrating significant in vivo effectiveness and cytokine release. However, this was not evaluated for the LRKK2 system, and the usage of IKK-16 is now restricted to IKK alone [116,117].

Estrada et al. found GNE0877 and GNE-9605 as highly effective, selective, and brain-penetrant LRRK2 inhibitors by selecting aminopyrazole-based hit compounds and applying structural changes [119]. GNE-0877 had cellular potency of 3 nM, but GNE-9605 had cellular potency of 18.7 nM. GNE-0877 had an excellent pharmacokinetic profile but was found to be a CYP1A2 reversible inhibitor. Furthermore, compound GNE-9605 was not shown to be a CYP isoform inhibitor. The compound was discovered to be extremely selective against 178 distinct kinases, inhibiting only 1 kinase more than 50% of the time. Both compounds were tested in vivo for suppression of S1292 phosphorylation at doses of 10 and 50 mg/kg. Pharmacokinetic investigations were also conducted on the Cynomolgus monkey, with positive findings. The author chose GNE-9605 for preclinical and genotoxicity testing.

PF-06447475 is a highly selective brain-penetrant pyridopyrimidines-based LRRK2 inhibitor discovered using a high throughput screening strategy [120]. In a biochemical study, the inhibitor had a low nanomolar affinity for WT LRRK2 and G2019S, with IC_50_ values of 3 and 11 nM, respectively. The compounds had reasonable pharmacokinetic characteristics, while clearance and oral bioavailability were moderate and low, respectively. Interestingly, compounds had no effect on Type II pneumocytes or impaired renal function in rats after 15 days of therapy. Daher et al. discovered a similar response in rats after a 4-week treatment with PF-06447475 [121]. Furthermore, the authors revealed that pyridopyrimidine scaffolds may make hydrogen bond interactions with LRRK2 hinge region residues E1948 and A1950. PF-06447475 has shown to be an effective tool for studying central and peripheral LRRK2 biology.

GNE-7915 was further modified to produce JH-II-127, which inhibited LRRK2 with IC_50_ values of 6.6 nM, 2.2 nM, and 47.7 nM for LRRK2-wild-type, G2019S, and A2016T, respectively. JH-II-127 is a brain-penetrant, highly effective, and selective LRRK2 inhibitor that strongly inhibits S910 and S935 phosphorylation of wild-type LRRK2 and G2019S in cells and can reduce S935 phosphorylation in San Pedro mouse brain following oral treatment at a low dosage of 30 mg/kg [122]. JH-II-127 had good oral bioavailability, half-life, and plasma exposure, according to the pharmacokinetic properties study. The inhibitor forms hydrogen bond interactions with hinge region residues M1949 and A1950, according to a molecular docking study using the Roco kinase structure. Although the inhibitor demonstrated high selectivity, the authors noted that further kinase prowling and in vivo pharmacokinetic studies were needed to make it an ideal LRRK2 inhibitor [122]. The emergence of the indazole category of LRRK2 inhibitors led to the discovery of MLi2. The in vitro kinase experiment performed against the G2019S mutant revealed that this protein was inhibited with an IC_50_ of 0.76 nM [123,124]. Furthermore, a cellular LRRK2 phosphorylation inhibition assay at S935 revealed an IC_50_ of 1.4 nM. In an in vitro kinase profile TR-FRET test, MLi-2 demonstrates over 295-fold selectivity for over 308 kinases. In mice, the pharmacokinetic profile of MLi2 shows high absorption and oral bioavailability. Despite having a significant pharmacologic impact, as evidenced by the inhibition of pSer935 LRRK2, MLi-2 could not stop the progressive motor phenotype in MitoPark and hence failed to reduce striatal dopamine, DOPAC, and tyrosine hydroxylase levels. Furthermore, the scientists indicated that extended MLi2 therapy may result in type II pneumocytes [123].

Pfizer developed PF-06685360 (PFE-360), a pyridopyrimidine scaffold-based inhibitor (Figure 4) [125]. PFE-360 is an orally active, potent, selective, brain-penetrating LRRK2 inhibitor with an in vivo IC_50_ of 2.3 nM [125,126]. Furthermore, the authors investigated the safety concerns made against LRRK2 inhibitors in peripheral organs such as the lung and kidney. It was shown that a lower dose of the inhibitor can minimize the lung effect. A recent study found that 12 weeks of oral PFE-360 administration at 7.5 mg/kg reduced LRRK2 activity in the kidney, resulting in morphological changes such as darker kidneys and a progressive accumulation of hyaline droplets in the renal tubular epithelium [126]. Furthermore, these morphological alterations were partially reversible after a 30-day treatment-free interval, indicating that pharmacological LRRK2 inhibition may not have a negative impact on kidney function.

Compound **7** (LRRK2 inhibitor 1) was discovered in a series of 7H-pyrrolo [2,3-d] pyrimidin-2-amine derivatives. In an homogeneous time-resolved fluorescence (HTRF) assay Compound **7** displayed an IC_50_ against LRRK2 of 6.8 nM [127]. Compound **7**’s kinase selectivity was investigated utilizing the HotSpot test platform comprising over 340 more kinases. It was demonstrated to be a very selective LRRK2 inhibitor, with just three off targets (ALK, IRR, and TSSK1) in the KINOMEscan profile. Compound **7** demonstrated excellent permeability, solubility, metabolic stability, and CNS penetration. The phosphorylation of LRRK2 Ser935 was reduced concentration-dependently, with pIC_50_ values of 7.5 in AHE and 7.8 in ANK. Compound **7**’s in vivo pharmacology was next evaluated by measuring inhibition of LRRK2 S935 phosphorylation in the brain, lung, spleen, and kidney after oral administration to rats at 10 mg/kg, 30 mg/kg, and 100 mg/kg as well as to mice (45 mg/kg). Compound **7** inhibited CYP3A4 significantly but had a lesser affinity for hERG. The authors concluded that further optimization of this series toward a clinical candidate would be published in due time [127].

EB-42486 was discovered after a high throughput screening effort against the G2019S variant (Figure 4). A single indazole-based hit candidate was chosen from 50,000 compounds and demonstrated substantial inhibition of G2019S LRRK2. The compound’s binding mode was further studied using the LRRK2 homology model. This knowledge was utilized to generate a succession of compounds from the hit compound and eventually led to the development of EB-42486 [128]. The compound had IC_50_ values of 6.6 and 0.2 nM against wild-type and G2019S variants, respectively. Furthermore, phosphorylation inhibition at S935 and S1292 was investigated using HEK293 cell lines and selectivity. At 100 nM concentration, the chemical showed high selectivity, inhibiting only 10 kinases. Unfortunately, the chemical had poor pharmacokinetic qualities in mice, with little brain exposure at a 2 mg/kg dosage. Merk released a study that found aminoquinazoline-derived hit compounds to be powerful, selective, and brain penetrant LRRK2 inhibitors. Following in vitro tests, the researchers used in silico approaches to address BBB and CNS-related obstacles. This led to an early hit with strong pharmacokinetic qualities; compounds were further modified to improve selectivity and pharmacokinetic profile, resulting in the identification of compounds **22** and **24** [129]. Compound **22** inhibited G2019S LRRK2 significantly in biochemical and cellular assays, with IC_50_ values of 0.6 nM and 0.57 nM, respectively. Compound **22** had a decent pharmacokinetic profile in rats; however, it had a considerable efflux in the P-gp test. Furthermore, the ex vivo PMBC test findings showed good unbound potency against wild-type LRRK2 with an IC_50_ of 1.9 nM, consistent with the SHSSY5Y cell line data. Compound **22** impacted the activity of just three off-targets in a thorough screen that included 263 and 118 additional kinases in in vitro experiments (ERK1 IC_50_ = 1.8 M, tachykinin NK1 IC_50_ = 5.7 M, and adenosine transporter IC_50_ = 9.6 M), each of which was affected at high dosages. In vivo pharmacodynamics showed that compound **22** inhibited pS935 in the rat striatum in a dose-dependent manner. Similar findings were achieved in dogs but not in rhesus monkeys. Furthermore, the compound was not shown to be mutagenic or genotoxic. Compound **22** was not chosen as the final lead compound due to the possible danger of disproportionate drug metabolite production; instead, compound **24** was designed. The compounds showed 2.6 nM selectivity against LRRK2 and only inhibited two kinases, MARK3 and ANF. In terms of thermodynamic solubility, compound **24** outperformed compound **22** (FaSSIF thermodynamic solubility = 0.03 mg/mL) while staying stable. The preclinical and clinical assessment of LRRK2 inhibitor DNL201, also named GNE-0877 in previous reports, was disclosed by Denali Therapeutics recently [130]. DNL201’s cellular potency was tested in HEK293 cells overexpressing LRRK2 G2019S utilizing S935 and S1292 LRRK2 phosphorylation as indicators of kinase activity. DNL201 displayed an IC_50_ of 47 nM for pS935 LRRK2 and 45 nM for pS1292 in HEK293 cells. The dose dependent effect was observed in pS935 LRRK2 and pT73 Rab10 in human pluripotent stem cell derived microglia, moreover similar results were also observed in CNS cells. Human PBMCs from LRRK2 G2019S carriers and controls were tested ex vivo for DNL201’s efficacy. Interestingly, DNL201 decreased LRRK2 phosphorylation by twofold. The treatment of DNL201 was also observed to restore the lysosome defects in neurons and astrocytes. The compound showed dose dependent inhibition of LRRK2 kinase activity in the periphery and CNS in animal models. Moreover, the researchers observed microscopic changes in lung tissues, but the effects were reversible and not found pathogenic. Therefore, compound was progressed to human clinical trials. The DNL201 phase I trial with 122 healthy volunteers as well as a Phase 1b study with 28 PD patients were completed. In the phase I and phase Ib clinical studies, DNL201 inhibited LRRK2 at single and repeated doses, demonstrating LRRK2 and lysosomal pathway engagement. Denali Therapeutics will shortly begin late-stage clinical studies for DNL201 and another inhibitor, DNL151 (BIIB122) [130].

## 7. LRRK2 Degraders

For decades, pharmaceutical research has focused on finding small molecules that, using occupancy-driven pharmacology, block the active or regulatory portions of receptors, limiting the activity of disease-causing proteins. Moreover, this approach is not found successful for a certain class of proteins due to poor selectivity and off-target effects. An excellent approach to overcome the limitations of small molecule–induced toxicities in certain cell types named as Antisense oligonucleotides (ASOs) was tested successfully on LRRK2 induced pathologies. ASOs are the short 10–20 bp long oligonucleotides which selectively target the expression of mRNA. ASOs can function through a variety of methods, such as by targeting mRNA for degradation by the cellular endonuclease ribonuclease H (RNase H) or by binding and inhibiting pre-mRNA splice sites to affect splicing and production of the final mRNA product [131,132]. The intracerebroventricular injections of ASOs can be delivered directly to brain without any carrier vesicle and therefore provide an advantage over small molecule–based therapies. In a mouse model of α-synuclein brain infection, LRRK2 ASOs reduced fibril development and dopaminergic neuron loss. A Phase 1 safety trial of ASO BIIB094 in PD patients is ongoing [133]. Korecka et al. recently reported splice-switching antisense oligonucleotides against LRRK2 WT or G2019S. According to this study a single intracerebroventricular injection of ASO induces exon-41 skipping which ultimately resulted in decreased Rab10 phosphorylation [134]. The ASO technology has made remarkable advances in past 15 years, with 10 FDA approvals against various disorders [135]. A few challenges still need to optimization in the areas of developing more conventional ASO delivery, target engagement, and the safety profile for clinical use. In recent years, academic institutions and pharmaceutical industries have paid close attention to targeted protein degradation, with the proteolysis targeting chimeras (PROTAC) technique [136]. PROTAC is a novel technique that employs small chemicals to stimulate protein breakdown in order to modulate protein levels [137,138]. As a novel drug development technique, its mechanism of action differs from that of standard small-molecule inhibitors, and it offers enormous promise for overcoming drug resistance and off-target effects. PROTACs are heterobifunctional molecules that incorporate an E3 ligase ligand, a linker, and a target protein ligand (Figure 5A). PROTACs induce the development of a target protein–E3 ligase complex, which eventually leads to the target protein’s destruction via the proteasome (Figure 5A). In the past decade, significant progress has been made in PROTAC research for finding suitable clinical candidates against a number of cancer-causing genes [139]. In neurodegenerative disorders (NDs), misfolded protein aggregates are difficult to remove using conventional drug discovery methods. In such situations, protein degradation methods such as PROTACs look promising for neurodegenerative disorder treatment, allowing selective removal of “undruggable” target proteins via physiological protein degradation [140]. Furthermore, developing the optimal combination of LRRK2 ligand (brain-penetrant and selective over WT-LRRK2) and E3 ubiquitin ligase ligand to accomplish selective and regulated protein degradation is a big challenge [61]. LRRK2 targeting inhibitor development–based studies have generally focused on Type I kinase inhibitors, which have a few limitations as suppressed protein might interfere with natural systems, potentially leading to undesired side effects [20]. However, most of the recently identified inhibitors displayed excellent selectivity and brain-penetrant potential and have opened a window for the development of desirable LRRK2-PROTAC [81]. Interestingly, according to the recent literature published, similar approaches were followed for the development of LRRK2-PROTAC. The researcher exploited Type I inhibitors to build an LRRK2 targeting PROTAC. The Dana-Farber Cancer Institute reported the first report on LRRK2 degraders [141]. The representative bifunctional compounds (degrader or PROTAC) contain a targeting ligand (an aminopyrimidine or indazole) that binds LRRK2, a degron that binds an E3 ubiquitin ligase (lenalidomide and pomalidomide), and a linker with a moiety that covalently connects the degron and the targeting ligand (Figure 5B) [142]. The researchers claimed that chemicals successfully block S935 phosphorylation with nanomolar efficacy and reduce the WT and G2019S LRRK2 levels. The two selected LRRK2 inhibitors, PF-06447475 and GNE-7915, were also exploited to develop cereblon targeting PROTACs. The synthesized PROTACs displayed potent kinase inhibition and the ability to penetrate cells, but, unfortunately, the degradation results obtained were not satisfactory [143]. These results indicate the challenges in the development of LRRK2 degraders after utilizing highly selective LRRK2 inhibitors. Recently Liu et discovered potent, selective, orally bioavailable brain penetrant LRRK2 degrader XL01126 (Figure 5C) [144]. In this work, the authors selected brain-penetrant LRRK2 inhibitor HG-10-102-01 as the LRRK2 binder. E3 ligase was selected after studying the effect of Cereblon (CRBN), cellular inhibitor of apoptosis (cAIP), and VHL with their corresponding ligands. After insertion of the linker of choice, the effect of the number of PROTACs was studied in mice and the level of pS935 was monitored. Based on degradation results, the authors designed second-generation PROTAC molecules by optimizing the LRRK2 ligand, linker, and VHL ligand, which ultimately resulted in the development of XL01126 [144]. The compound displayed significant selectivity and brain-penetration potential, although it violated Lipinski’s rule of five. The compound also displayed poor in vitro ADME properties; further pharmacokinetic and dynamics studies are needed to make this an ideal LRRK2 degrader. The application of PROTAC in neurodegenerative disorders is at the initial stage, but this technique is well established in undruggable cancer targets. For the successful application of PROTAC in neurovegetative disease, certain issues related to the toxicity of PROTACs due to off-target effects, pharmacokinetic properties, and blood–brain permeability must be addressed [145].

## 8. Conclusions and Future Perspective

LRRK2, a huge and complex protein, has been linked to the etiology of PD. Mutations in the GTPase and Kinase domains of LRRK2 can cause disease. The GTPase and kinase domains of LRRK2 are engaged in intricate regulatory interactions [80,99]. The protein has the ability to interact with a variety of essential signaling pathways in the cell [37]. Though the studies discussed in this review add to a better understanding of LRRK2’s activities and roles in Parkinson’s disease, the underlying processes remain unknown and require more investigation. According to current understanding, multiple LRRK2 inhibitors have been developed by academic and industry organizations, each with improved potency, selectivity, and brain penetrability. LRRK2 inhibitors may thus be used in clinical practice to treat Parkinson’s disease in the future. However, their effectiveness and safety remain concerns that must be addressed before they may be employed. The most important unresolved medicinal chemistry challenge is the lack of a crystal structure of the LRRK2 kinase domain. However, recent developments in low-resolution, cryo-EM-based LRRK2 structures provide an option for structure-based drug design, which was previously restricted to homology model–based techniques [24,56]. Utilizing the power of structure-based drug design methodologies such as pharmacophore modeling and quantitative-structure activity relationship (QSAR), highly selective inhibitors can be identified for WT as well as pathogenic variants. Tan et al. recently applied the molecular modeling approach on selected LRRK2 Type I inhibitors to study the detailed binding mode between the protein and ligand. The molecular dynamics simulation results reveal the key amino acids Glu1948 and Ala1950 are responsible for intermolecular hydrogen bond interactions between LRRK2 G2019S and inhibitor compounds [70]. Recent development of LRRK2 degraders using ASOs and Type I LRRK2 inhibitor scaffolds gives another hope to developing novel LRRK2 degraders which can overcome the limitations of Type I inhibitors [109,116,141,142,143,144]. Since LRRK2 is involved in multiple important cellular functions, its degradation may cause PD, according to a recent report. However, a study conducted by Blauwendraat et. reveals that haploinsufficiency of LRRK2 is neither a cause nor a protector of PD [146]. Therefore, kinase inhibition of WT and mutant variants remain a viable therapeutic approach. The next significant issue for the field is to find reliable biomarkers for detecting LRRK2 activity and follow the progression of Parkinson’s disease from the beginning. The LRRK2 GTPase domain, which is also linked with PD-causing mutations, has been identified as a viable alternative to the kinase domain. However, very few small molecules that inhibit GTPase activity have been reported to date; therefore, these can be explored in future studies [147]. Furthermore, targeting allosteric sites, blocking protein–protein interactions, connection with Rab proteins, and LRRK2 dimerization, among other strategies, may be effective in inhibiting LRRK2 and require further studies in this area. An interesting approach proposed by Berndsen et al. is the dephosphorylation of the Rab protein by increasing the activity of PPM1H phosphatase [148]. By modulating the dephosphorylation of Rab proteins, PPM1H functions as a crucial regulator of LRRK2 signaling. Therefore, further research can be designed in the field of PPM1H activity enhancers to prevent or cure LRRK2-induced PD [148,149].

## Figures and Tables

**Figure 1 genes-13-01426-f001:**
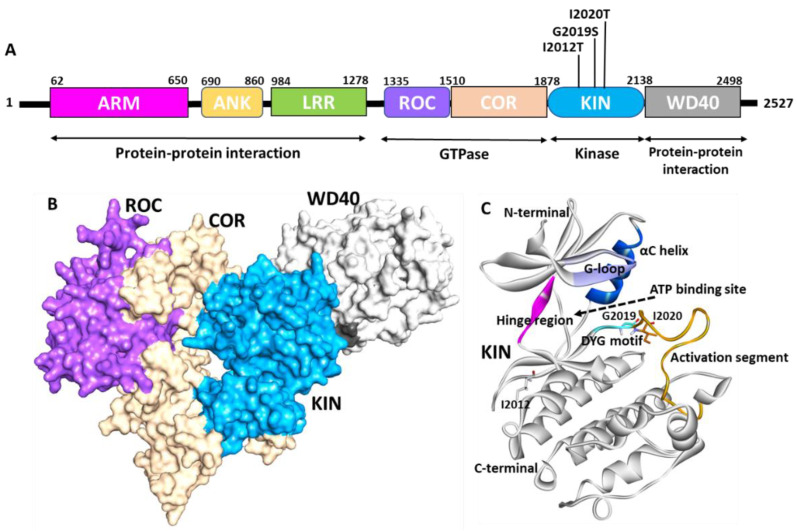
Structure of LRRK2. (**A**) Domain organization of LRRK2. ARM—Armadillo repeats (pink), ANK—ankyrin repeats (yellow), LRR—leucine-rich repeat (green), ROC—Ras of complex (purple), COR—C terminal of Roc (cream), KIN—kinase domain (blue), WD40—WD repeat domain (grey). The mutation of the kinase domain is highlighted in black. The function of each domain is marked below the 2D diagram. (**B**) Surface representation of the cryo-EM structure of LRRK2. The color code was used the same as shown in the domain diagram. (**C**) Kinase domain of LRRK2. Active site region G-loop (purple), hinge (pink), DYG motif (cyan), αC-helix (blue), activation segment (orange) are highlighted. The position of mutation associated with PD is shown by stick representation. Figure created using Discovery Studio (DS) v19 (accessed on 10 February 2022) [55].

**Figure 2 genes-13-01426-f002:**
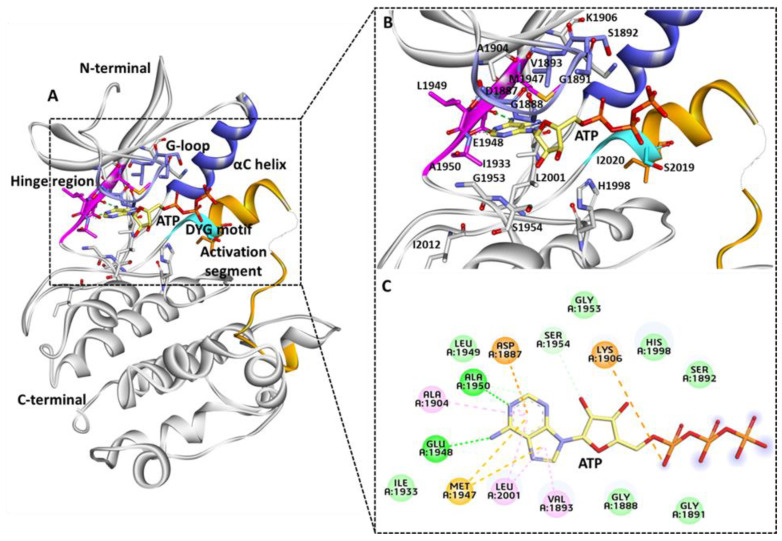
(**A**) Kinase domain ATP binding site. (**B**,**C**) Key molecular interactions involved in ATP interaction 3D and 2D representation. Important regions of active site G-loop (purple), hinge region (pink), DYG motif (cyan), and activation segment (orange), αC helix (blue), and the remaining part of the LRRK2 kinase domain is shown in grey-colored ribbon. ATP molecule is shown with a yellow stick. Figure created using Discovery Studio (DS) v19 (accessed on 12 February 2022) [69].

**Figure 3 genes-13-01426-f003:**
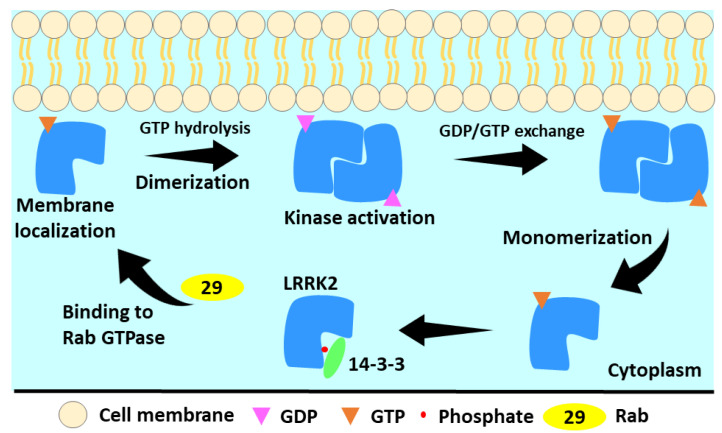
The LRRK2 activation cycle. In the cytosol, LRRK2 is monomeric and GTP-bound. This less active state forms a cytoplasmic complex with 14-3-3 proteins. LRRK2 is membrane-localized by activated Rab proteins. GTP hydrolysis occurs at membrane and results in LRRK2 dimerization. During the GTPase cycle, LRRK2 phosphorylates its substrates. Low LRRK2 affinity for GDP allows quick GDP release, rebinding of GTP, and monomerization of LRRK2 [100].

**Figure 4 genes-13-01426-f004:**
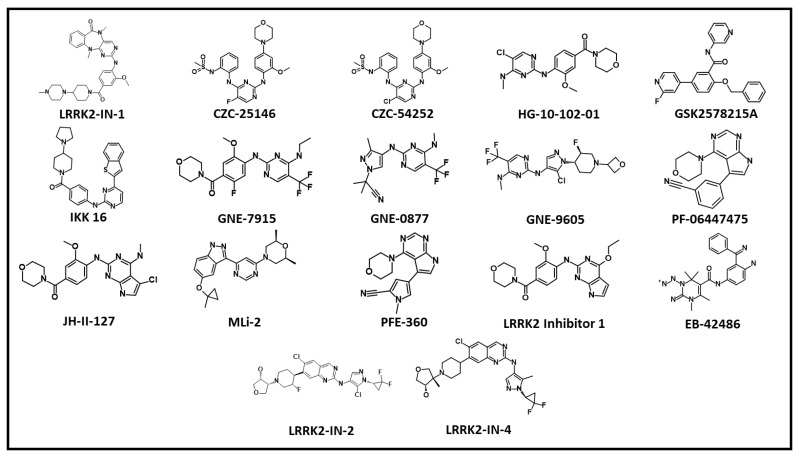
The 2D chemical structures of ATP competitive LRRK2 inhibitors.

**Figure 5 genes-13-01426-f005:**
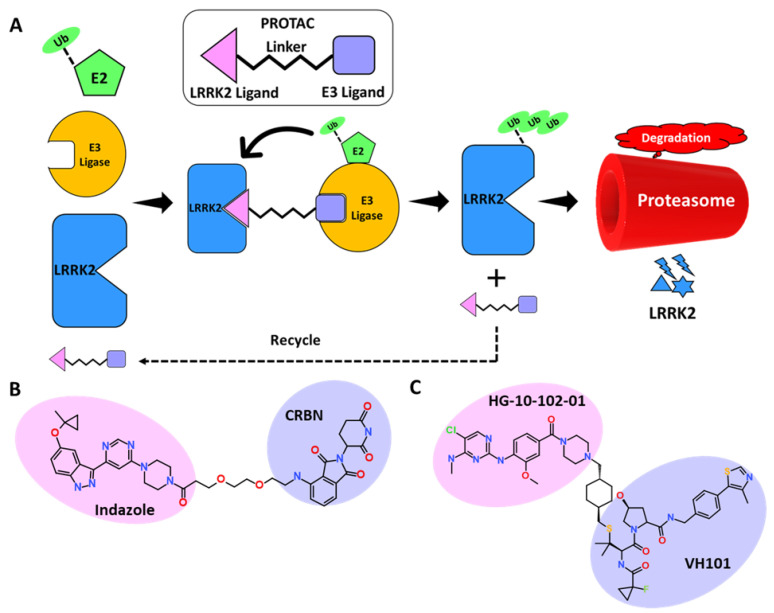
(**A**) Schematic representation of the assembly of PROTAC machinery. (**B**,**C**) Representative PROTAC developed against LRRK2.

**Table 1 genes-13-01426-t001:** The inhibitory activity of selected LRRK2 inhibitors against WT and G2019S, kinase selectivity, and brain penetration potential are summarized.

LRKK2 Inhibitor	Variants		Kinase Selectivity	Brain Penetration	References
	WT	G2019S			
LRRK2-IN-1	13	6	Yes	No	[112]
CZC-25146	4.76	6.87	Yes	No	[113]
CZC-54252	1.28	1.85	Yes	No	
HG-10-102-01	20.3	3.2	Yes	Yes	[114]
GSK2578215A	10.9	8.9	Yes	No	[115]
IKK 16	50	-	No	No	[116,117]
GNE-7915	9	-	Yes	Yes	[118]
GNE-0877	3	-	Yes	Yes	[119]
GNE-9605	19	-	Yes	Yes	
PF-06447475	3	11	Yes	Yes	[120,121]
JH-II-127	6.6	2.2	Yes	Yes	[122]
MLi-2	-	0.76	Yes	Yes	[123,124]
PFE-360	2.3	-	Yes	Yes	[125,126]
LRRK2 Inhibitor 1	6.8	-	Yes	Yes	[127]
EB-42486	6.6	0.2	Yes	Yes	[128]
Compound **22**	-	0.6	Yes	Yes	[129]
Compound **24**	-	2.6	Yes	Yes	

## Data Availability

Not applicable.

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
