# Peer review of "Structural Insights and Development of LRRK2 Inhibitors for Parkinson’s Disease in the Last Decade"

_genes, 2022, doi:10.3390/genes13081426_

Round 1

Reviewer 1 Report

This review provides valuable insights into the development of inhibitors for LRRK2 from the structure point of view. It is well written and organized. However, part of the content requires revision before being accepted for publication.

Comments:

1.      I suggest to add one more figure to illustrated the activation of kinase domain. It’s better to contain the structure information as the title says “structural insight”.

2.      Please combine figure 3 and 4 since they all show the inhibitors and even have the same title.

3.      The authors gave very detailed information about the inhibitors in the section 6 “Kinase inhibitors”, however, it also makes it too lengthy. And for some inhibitors, they have been discussed elsewhere in details, for instance in the review, Wojewska, D.N.; Kortholt, A. Biomolecules 2021, 11, 1101. https://doi.org/10.3390/biom11081101. Please try to shorten this part and focus more on recent developments and more related to the structure.

4.      Line 109-110, please add corresponding references for this sentence.

Author Response

Reviewer 1

Comments and Suggestions for Authors

This review provides valuable insights into the development of inhibitors for LRRK2 from the structure point of view. It is well written and organized. However, part of the content requires revision before being accepted for publication.

We are thankful to the reviewer for reviewing our manuscript and providing important insights to improve the quality of the manuscript. As per suggestions, we have modified the manuscript and the pointwise description was provided below.

 Comments:

1) I suggest to add one more figure to illustrate the activation of kinase domain. It’s better to contain the structure information as the title says “structural insight”.

Justification: We have considered the reviewer suggestion and inserted a Figure for the activation of kinase (Figure 3) on page number 7

2) Please combine figure 3 and 4 since they all show the inhibitors and even have the same title.

Justification: As per suggestion Figures 3 and 4 are combined and named Figure 4 shown on page number 11.

3) The authors gave very detailed information about the inhibitors in the section 6 “Kinase inhibitors”, however, it also makes it too lengthy. And for some inhibitors, they have been discussed elsewhere in details, for instance in the review, Wojewska, D.N.; Kortholt, A. Biomolecules 2021, 11, 1101. https://doi.org/10.3390/biom11081101. Please try to shorten this part and focus more on recent developments and more related to the structure.

Justification: Respected reviewer, we have gone through the following publication. Researchers discussed a few inhibitors which we have also discussed, they have not discussed the chemical and biological properties of the compound. Moreover, as per your suggestion, we reduced the discussion to important information only. The changes were highlighted in the section “Kinase Inhibitors” with red color fonts. Moreover, recent information on LRRK2 inhibitors has been added (line number 466-484)

4) Line 109-110, please add corresponding references for this sentence.

Justification: Citation has been provided on line number 115.

Reviewer 2 Report

This manuscript has nicely described the state-of-the-art of targeting LRRK2 kinase activity as potential therapeutic approach for Parkinson’s disease (PD). The authors have provided clear overviews and detailed descriptions of LRRK2 structure and biology, which support the prominent role of LRRK2 PD pathogenesis and justifies the development of LRRK2 inhibitors or LRRK2 degraders as promising disease-modifying treatments for PD.

I found that the manuscript was very well structured, nicely written with appropriate illustrations and tables. As LRRK2 inhibitors are currently being evaluated in clinical trials, this review article offers an useful reference for the field summarizing what has been done and what could be expected in a near future.  

Here are my suggestions to improve the manuscript:

1)     Typo mistakes or inconsistent wordings:

a.      Abstract, line 12: I believe Parkinson’s disease is the second most prevalent neurodegenerative disease, but not the most prevalent

b.      Introduction, line 31: I am not sure we can say “the medicine levodopa”? or would that be better to say “L-Dopa replacement

c.      Introduction, line 58: “Variations in the aforementioned genes are the most prevalent causes of PD”. I think the authors may want to say “the most prevalent causes of familial PD” here, as most PD cases are sporadic.

d.      Minor typo errors across the manuscript:  Please be consistent between Type 1 or Type1, SH-SY5Y or SHSY5Y, HEK-293 or HEK293, GNE-0877 or GNE0877

2)     Chapter 2: LRRK2 and its Function. The authors may need to consider extending a little more the description of the enzymatic function of LRRK2, especially the kinase function. It may worth it to mention findings about LRRK2 kinase substrates in vitro vs in cells (autophosphorylation at Ser1292 or phosphorylation a subset of Rab proteins at conserved residues of Switch II motif) (Islam and Moore, 2017; Alessi and Sammler, 2018; Taylor and Alessi, 2020).  It would be important to extent the discussion on the potential implication of Rab phosphorylation in PD pathogenesis regarding recent findings on LRRK2-mediated Rab phosphorylation in cells and in vivo (Steger et al., 2016; Steger et al., 2017; Dhekne et al., 2018; Bersden et al., 2019; Khan et al., 2021; Malik et al., 2021)

3)     Chapter 4: LRRK2 Mutations. Please consider updating the literature for LRRK2 mutations. Although most assays for LRRK2 functions were done with confirmed pathogenic mutations of LRRK2, these mutations are rather rare. Other more common mutations also been described in literature to alter PD risk. Some of them (N551K, R1398H, G2385R…) have been described to modify LRRK2 activity (Rudenko et al., 2012; Nixon-Abell et al., 2016; Hui et al., 2018; Wang et al., 2020; Zhang et al., 2021; Ordonez et al., 2022). It is also important to mention that although in vitro kinase activity may be variable between pathogenic LRRK2 mutants, they consistently increase Rab phosphorylation in cells, highlighting the role of Rab phosphorylation in LRRK2 biology.

4)     Chapter 6: Kinase Inhibitors. This is a long chapter and also the main body of the article. The authors have done a nice job summarizing pharmacological properties of different LRRK2 inhibitors. However, the text was rather long and mainly listed the literature. The authors could consider to extent the summary table (Table 1) to include some of the key pharmacological properties of these compounds to help the readers.  Additionally, it appears important to my mind that the authors clarify what are different types of LRRK2 inhibitors, how are they differ from each other and what could be done to improve their efficacy. Recent updates on this topic may need to be included (Lesniak et al., 2022; Jennings et al., 2022; Kingwell, 2022).

5)     Chapter 7: LRRK2 degraders. Please discuss the potential effects of complete, whole-body removal LRRK2 in human (Blauwendraat et al., 2018). Is PROTAC the only way to removal LRRK2? What is the advantage of PROTAC vs ASO or other approaches downregulating LRRK2?

Author Response

Reviewer 2

This manuscript has nicely described the state-of-the-art of targeting LRRK2 kinase activity as potential therapeutic approach for Parkinson’s disease (PD). The authors have provided clear overviews and detailed descriptions of LRRK2 structure and biology, which support the prominent role of LRRK2 PD pathogenesis and justifies the development of LRRK2 inhibitors or LRRK2 degraders as promising disease-modifying treatments for PD.

I found that the manuscript was very well structured, nicely written with appropriate illustrations and tables. As LRRK2 inhibitors are currently being evaluated in clinical trials, this review article offers an useful reference for the field summarizing what has been done and what could be expected in a near future.  

The authors are very thankful to the reviewer for a detailed review of each section of the manuscript. All the suggestions are valuable and therefore we tried to improve the manuscript accordingly. All the changes were highlighted in the manuscript in RED color and brief details were provided below.

Here are my suggestions to improve the manuscript:

1)     Typo mistakes or inconsistent wordings:

  1. Abstract, line 12: I believe Parkinson’s disease is the secondmost prevalent neurodegenerative disease, but not the most prevalent

Justification: We agree with the reviewer and updated line 12 as per suggestion.

  1. Introduction, line 31: I am not sure we can say “the medicine levodopa”? or would that be better to say “L-Dopa replacement

Justification: Yes we agree with the reviewer and changed the terminology used to “dopamine replacement using L-dehydroxyphenylalanine (L-DOPA)”

  1. Introduction, line 58: “Variations in the aforementioned genes are the most prevalent causes of PD”. I think the authors may want to say “the most prevalent causes of familial PD” here, as most PD cases are sporadic.

Justification: We corrected the line 58 to familial PD

  1. Minor typo errors across the manuscript:  Please be consistent between Type 1or Type1SH-SY5Y or SHSY5YHEK-293 or HEK293, GNE-0877 or GNE0877

Justification: We have uniformed the pattern for these special words.

2)     Chapter 2: LRRK2 and its Function. The authors may need to consider extending a little more the description of the enzymatic function of LRRK2, especially the kinase function. It may worth it to mention findings about LRRK2 kinase substrates in vitro vs in cells (autophosphorylation at Ser1292 or phosphorylation a subset of Rab proteins at conserved residues of Switch II motif) (Islam and Moore, 2017; Alessi and Sammler, 2018; Taylor and Alessi, 2020).  It would be important to extent the discussion on the potential implication of Rab phosphorylation in PD pathogenesis regarding recent findings on LRRK2-mediated Rab phosphorylation in cells and in vivo (Steger et al., 2016; Steger et al., 2017; Dhekne et al., 2018; Bersden et al., 2019; Khan et al., 2021; Malik et al., 2021)

     Justification: Respected reviewer we are thankful for the valuable suggestion the suggested publications were studied and related content was summarized in section “LRRK2 and its function”. Line number 94-101 (REF cited 38-40:  Steger et al., 2016, Dhekne et al., 2018, Alessi and Sammler, 2018). Some of the work is summarized in section “Kinase activation” line number: 243-253 (REF cited 97: Sheng et al., 2012). 

      Few suggested publications were summarized in the “Conclusion section”. Line number: 602-606) REF cited 147, 148: Berndsen et al., 2019, Khan et al., 2021)

3)     Chapter 4: LRRK2 Mutations. Please consider updating the literature for LRRK2 mutations. Although most assays for LRRK2 functions were done with confirmed pathogenic mutations of LRRK2, these mutations are rather rare. Other more common mutations also been described in literature to alter PD risk. Some of them (N551K, R1398H, G2385R…) have been described to modify LRRK2 activity (Rudenko et al., 2012; Nixon-Abell et al., 2016; Hui et al., 2018; Wang et al., 2020; Zhang et al., 2021; Ordonez et al., 2022). It is also important to mention that although in vitro kinase activity may be variable between pathogenic LRRK2 mutants, they consistently increase Rab phosphorylation in cells, highlighting the role of Rab phosphorylation in LRRK2 biology.

      Justification: Suggested publications were reviewed and summarized in the manuscript. Line number 214-220 (REF cited: Rudenko et al., 2012, Lorenzo-Betancor et al., 2012; Gopalai et al., 2014; Nixon-Abell et al., 2016; Zheng et al., 2011; Lesage et al., 2008.

4)     Chapter 6: Kinase Inhibitors. This is a long chapter and also the main body of the article. The authors have done a nice job summarizing pharmacological properties of different LRRK2 inhibitors. However, the text was rather long and mainly listed the literature. The authors could consider to extent the summary table (Table 1) to include some of the key pharmacological properties of these compounds to help the readers.  Additionally, it appears important to my mind that the authors clarify what are different types of LRRK2 inhibitors, how are they differ from each other and what could be done to improve their efficacy. Recent updates on this topic may need to be included (Lesniak et al., 2022; Jennings et al., 2022; Kingwell, 2022).

     Justification: Respected reviewer, some of the important features we already added to Table 1,  Moreover as per the other reviewer’s suggestion we reduced the information provided.  We have mentioned that inhibitors considered are Type 1 ATP-competitive inhibitors also the information about their scaffold and discovery methods are provided in the text. Few methods which can be utilized to improve the inhibitor scaffolds were mentioned on line number: 580-587.

      Utilizing the power of structure-based drug design methodologies such as pharmacophore modeling and quantitative-structure activity relationship (QSAR) highly selective inhibitors can be identified for WT as well as pathogenic variants. Tan et al. recently applied a molecular modeling approach on selected LRRK2 Type I inhibitors to study the detailed binding mode between the protein and ligand. The molecular dynamics simulations results reveal the key amino acid Glu1948 and Ala1950 are responsible for intermolecular hydrogen bond interactions between LRRK2 G2019S and inhibitor compounds [69]

     Recent updates on this topic has been added using suggested publications Line number: 466-484  (REF cited 129, Jennings et al., 2022)

5)     Chapter 7: LRRK2 degraders. Please discuss the potential effects of complete, whole-body removal LRRK2 in human (Blauwendraat et al., 2018). Is PROTAC the only way to removal LRRK2? What is the advantage of PROTAC vs ASO or other approaches downregulating LRRK2?

      Justification: Respected reviewer, our goal was to consider small-molecule-based compounds but as per suggestion, we summarized a few important points about another powerful approach Antisense oligonucleotides (ASOs). ASOs can function through a variety of methods, such as by targeting mRNA for degradation by the cellular endonuclease ribonuclease H (RNase H) or by binding and inhibiting pre-mRNA splice sites to affect splicing and production of the final mRNA product. Brief detail about ASO work on LRRK2 is also incorporated in the manuscript. Line number: 489-506.

      Detailed analysis reveals that ASO technology is already in use for the splicing or degradation of LRRK2 and progressed to clinical trials. On the other hand, PROTACs designed for neurodegenerative disease or against LRRK2 are a growing area of research. PROTAC research may be challenging in this field but rational design can provide effective treatment options (Explained in manuscript: line number 518-521 & Line number: 555-559).

      Since the LRRK2 is involved in multiple important cellular functions, its degradation may cause PD according to a recent report. However, a study conducted by Blauwendraat et. reveals that the haploinsufficiency of LRRK2 is neither a cause nor a protector of PD. Therefore, kinase inhibition of WT and mutant variants remain a viable therapeutic approach. Line number: 590-594, REF cited 145, Blauwendraat et al, 2018)

Reviewer 3 Report

This review by Thakur and colleagues gives a nice overview of LRRK2, Parkinson’s disease and the current inhibitors in development. There are just a couple of suggestions below to improve this review.

-       I think the review would benefit from more information on the use of these compounds in animal models. The authors have described many studies related to pharmacokinetic properties but have not included many investigations using animal models of disease or at least addressed why there might be a paucity of such studies.

-       Minor: some sections could do with some improvement in grammar.

Author Response

Reviewer 3

This review by Thakur and colleagues gives a nice overview of LRRK2, Parkinson’s disease and the current inhibitors in development. There are just a couple of suggestions below to improve this review.

Respected reviewer, we are thankful for reviewing our manuscript and providing valuable suggestions for improvement of the manuscript. The manuscript was revised and changes were highlighted in RED in the manuscript.

1) I think the review would benefit from more information on the use of these compounds in animal models. The authors have described many studies related to pharmacokinetic properties but have not included many investigations using animal models of disease or at least addressed why there might be a paucity of such studies.

Justification: Respected reviewer our manuscript is focused on the development of Type 1 inhibitors against LRRK2. In the present manuscript, well-established commercially available inhibitors were selected. Some of the inhibitors lack animal studies but where ever the details on animal models were available we tried to include them in the section “Kinase Inhibitors” for example inhibitor MLi2, PF-06447475, PFE360, GNE-9606, EB-42486, Compound22, and clinical candidate GNE-0877 or DNL20 were discussed in section.

2) Minor: some sections could do with some improvement in grammar.

Justification: We thoroughly checked the manuscript for grammatical errors.

Round 2

Reviewer 1 Report

I'm satisfied with the response and revision made by the authors and my concerns are well addressed. Therefore, I suggest the publication of this manuscript.